# Diversity-oriented Deep Multi-modal Clustering

**Yanzheng Wang**[#], **Xin Yang**[#], **Yujun Wang**, **Shizhe Hu**[*], **Mingliang Xu**[*]

School of Computer and Artificial Intelligence, Zhengzhou University, China

## Abstract

Deep multi-modal clustering (DMC) aims to explore the correlated information from different modalities to improve the clustering performance. Most existing DMCs attempt to investigate the consistency or/and complementarity information by fusing all modalities, but this will lead to the following challenges: 1) Information conflicts between modalities emerge. 2) Information-rich modalities may be weakened. To address the above challenges, we propose a diversity-oriented deep multi-modal clustering (DDMC) method, where the core is dominant modality enhancement instead of multi-modal fusion. Specifically, we select the modality with the highest average silhouette coefficient as the dominant modality, then learn the diversity information between the dominant madality and the remaining ones with diversity learning, and finally enhance the dominant modality for clustering. Extensive experiments show the superiority of the proposed method over several compared DMC methods. To our knowledge, this is the first work to perform multi-modal clustering by enhancing the dominant modality instead of fusion.

## 1 Introduction

Deep multi-modal clustering (DMC) aims to integrate data from multiple modalities (e.g., image, text, audio, etc.) to classify the data through unsupervised learning. DMC combines the feature extraction ability of deep learning and the clustering idea of unsupervised learning, which is an important direction of multi-modal data analysis at present, and has achieved excellent performance in many fields, such as the medical field [1–3], autonomous driving and intelligent transportation [4, 5], and recommendation systems [6–8].

**Related Works.** At present, almost all DMCs cluster by fusion method. Based on different fusion stages, DMCs can be roughly divided into the following three categories: 1) Feature-level fusion: these methods [9–13] extract the features of different modalities and connect them into a single high-dimensional feature vector, which is used as a single input for clustering or feature learning. For example: Zhou and Shen [10] propose a method that first use an adversarial regularizer to align modalities, and then perform an attention fusion on all modalities, so as to quantify the importance of different modalities. 2) Decision-level fusion: these methods [14–16] first obtain independent modeling of each modality, obtain their own output or clustering results, and then fuse these results in the final stage. For example, Meng et al.[14] conduct the features of different modalities in their respective clustering layers for depression regression training, and linearly weighted sums the prediction results of each modality to obtain the fusion result as the final output result. 3) Mixed-level fusion: these methods [17–19] combine the output of a single mode prediction through feature-level fusion. For example, Morales et al.[17] train a separate model for each modality, then get the predictions for each modality, and finally train a new model on these new vectors to output the final prediction.

---

[#]Equal contribution
[*]Corresponding author (ieshizhehu@gmail.com)

**Motivations.** Although the above methods have achieved excellent performance, due to the use of different levels of fusion, the following challenges arise: 1): Inconsistency of multi-modal data: different modalities may provide conflicting or inconsistent information about the same thing. 2): Weight assignment and modalities importance: modalities fusion can cause some informative high-quality modalities to be forced to align with low-quality high-noise modalities [20]. To address the above challenges, we drew inspiration from [21]. As shown in Figure 1(a), people perceive the world in the process of the world about $83\%$ comes from the vision, and the remaining ways (such as smell, touch, taste and hearing) add up to only 17%. However, if the information obtained by the other ways is supplemented with the information obtained by the visual way, a more complete understanding will be obtained. Taking driving as an example, most of the information on the road condition comes from the vision, if at the same time through the radio, speakers and other information, we will have a more correct judgment on the road condition. Inspired by this, we propose a new multi-modal clustering method in Figure 1(b), in which modality 3 is the dominant modality. The diversity learning approach is employed to extract the diversity information between the dominant modality and the remaining modalities (modality 1, modality 2) and then the diversity information is spliced to the dominant modality for enhancement. Finally, the enhanced dominant modality is clustered.

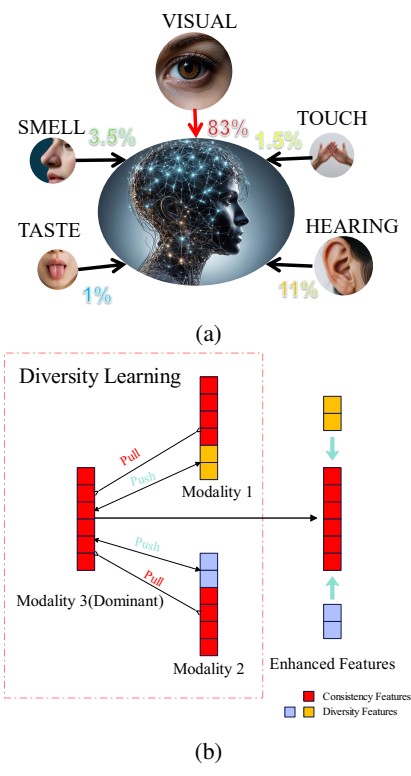

(a)

(b)

Figure 1: (a) The way humans perceive the world and its proportion. (b) The general idea of our method. Inspired by multi-modal assisted vision in human perception of the world.

**Contributions.** In this paper, we propose a diversity-oriented deep multi-modal clustering (DDMC) method. Specifically, our innovation lies in proposing a dominant modality enhancement strategy that 'enhancement instead of fusion'. In this method, we first select an informative modality as the dominant modality, and subsequently employ diversity learning to extract the diversity information from other modalities from the dominant modality. Finally, we concatenate these dissimilarity information to the dominant modality and perform the final clustering output through the clustering module. Compared with the latest DMCs method, our method achieves significant performance improvement on benchmark datasets, which verifies its effectiveness and advantages. The main contributions of our work can be summarised as follows:

- Ours is the first work to investigate multi-modal clustering by enhancing dominant modalities rather than fusing modalities.

- We propose a diversity-oriented deep multi-modal clustering method by dominant modality enhancement rather than modality fusion, which can maximally retain important information in the raw modality and has the advantages of both single-modal and multi-modal clustering.

- We can effectively enhance the dominant modality by simultaneously mining the diversity information of the remaining modality relative to the dominant modality at both the feature-level and the cluster-level. The results on multiple multi-modal datasets can validate the effectiveness of DDMC.

## 2 Methodology

A multi-modal dataset $X = \{X^1, \ldots, X^m, \ldots, X^M\}$ contains $N$ samples of $M$ modalities, where $X^m \in \mathbb{R}^{N \times d_m}$ denotes the samples of dimension $d_m$ from the $m$-th modality. Our purpose is to correctly divide $N$ samples into $K$ clusters by learning information between multiple modalities. To provide a more intuitive overview of the proposed method, we present it in Figure 2. Taking three modalities as an example, the dominant modality (modality 2 is shown in the figure) is first determined through the selection of dominant modality module. Subsequently, each modality is

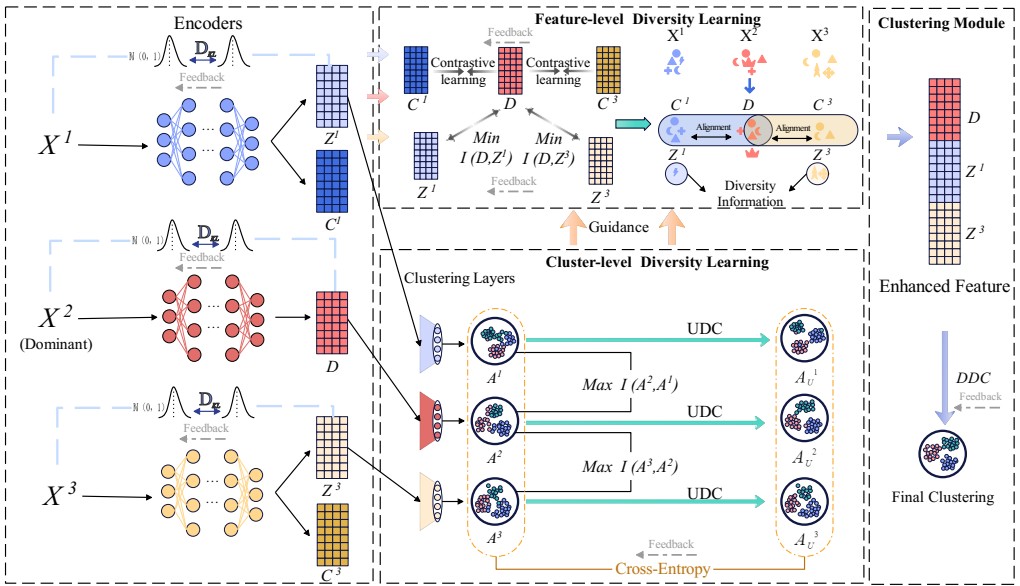

Figure 2: Details of the proposed framework. Firstly, the dominant modality feature $\mathbf{D}^{dm}$, consistency features $\{\mathbf{C}^m\}_{m \neq dm}^M$ and diversity features $\{\mathbf{Z}^m\}_{m \neq dm}^M$ are obtained through modality specific encoders. Then, through feature-level diversity learning and cluster-level diversity learning, the diversity information of the remaining modality is learned. Finally, the diversity information is enhanced for the dominant modality, and the final clustering is completed through the clustering module. Among them, UDC is Uniform Distribution Constraint, which forces the edge distribution of each cluster to be as close to uniform as possible. DDC is Deep Divergence-based Clustering.

modeled with its original features through independent variational encoders. Next, diversity learning is performed at both the feature-level and the cluster-level, and the diversity information obtained from the non-dominant modalities is used to enhance the representation of the dominant modality. Finally, the final clustering is performed based on the enhanced dominant modality features to obtain the clustering results.

## 2.1 Motivation: Modality Fusion Leads to Lower Clustering Efficiency

In order to learn the information between modality, researchers capture the connections of heterogeneous modality by fusing each modality. However, this also brings corresponding problems: unfair weights in the fusion can lead to the degradation of the results. Specifically, we illustrate it on the popular multi-modal dataset Flickr [22], as shown in Figure 3. We conducted 20 iterations of $K$-means [23] clustering for each modality and the fusion modality, representing the quality of individual modality and the fusion modality through its unsupervised accuracy. As shown in the figure, it can be clearly seen that modality 3 belongs to the high-quality modality, while modality 1 and 2 belong to the low-quality modality. Due to the unequal relationship between the weights allocated to each modality and the quality in the fusion, the clustering effect of the fused modality is worse. However, these low-quality

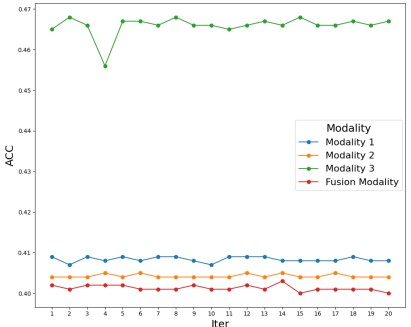

Figure 3: The accuracy rates of each modality and the fused modality after $K$-means clustering.

modality are not completely useless, we hope to select a high-quality modality as the dominant modality, and then learn useful information from the remaining modality to enhance the dominant modality. Therefore, unfair alignment of high-quality modality to low-quality modality can be avoided.

## 2.2 Selection of Dominant Modality

We decide that the dominant modality method is prior knowledge or the Silhouette coefficient (SI) fraction. Prior knowledge is that the modality weights in the dataset have been measured in previous work, or the importance of certain modality in some datasets is obvious. For these important modalities, the dominant modality is selected. For example, the importance weights of each modality are clearly given in [10]. In the absence of reliable prior information, the SI fraction is adopted to determine the dominant modality. SI is a metric used to evaluate the quality of clustering results. It combines the intra-cluster closeness and inter-cluster separation, and calculates a SI for each data point to measure its similarity to its own cluster and its nearest neighbor cluster. In the raw features, the clustering results are calculated by the $K$-means method, and then the SI score of each modality can be derived. For each sample $x_i^m$, the average SI of each modality is calculated as follows:

$$SI\left(\boldsymbol{X}^m\right) = \frac{1}{N} \sum_{\boldsymbol{x}_i^m \in \boldsymbol{X}^m} SI\left(\boldsymbol{x}_i^m\right), \tag{1}$$

where:

$$SI\left(\mathbf{x}_i^m\right) = \frac{b\left(\boldsymbol{x}_i^m\right) - a\left(\boldsymbol{x}_i^m\right)}{\max\left\{b\left(\boldsymbol{x}_i^m\right), a\left(\boldsymbol{x}_i^m\right)\right\}}, \tag{2}$$

where $a\left(\boldsymbol{x}_i^m\right) = \frac{1}{|C_k^m|} \sum_{\boldsymbol{x}_j^m \in C_k^m, \boldsymbol{x}_i^m \neq \boldsymbol{x}_j^m} d\left(\boldsymbol{x}_i^m, \boldsymbol{x}_j^m\right)$ is to calculate the average distance between this sample point and all other points in the same cluster, and $b\left(\boldsymbol{x}_i^m\right) = \min_{l \in [1,K], l \neq k} \frac{1}{|C_l^m|} \sum_{\boldsymbol{x}_j^m \in C_l^m} d\left(\boldsymbol{x}_i^m, \boldsymbol{x}_j^m\right)$ is to calculate the average distance between this sample point and all points in the nearest neighboring cluster. $|C_l^m|$ is the number of all sample points in the cluster where $x$ is located, and $d(,)$ is the distance between the two sample points.

The modality with the highest SI is the dominant modality $\mathbf{X}^{dm}$ is given by:

$$X^{dm} = \max \quad \{SI(X^1), SI(X^2), ..., SI(X^M)\}. \tag{3}$$

## 2.3 Diversity Learning

After selecting the dominant modality, each modality is encoded by an independent variational encoder. The dominant modality generates feature $\mathbf{D}^{dm}$, and the remaining modalities generate two sets of features, diversity features $\{\mathbf{Z}^m\}_{m \neq dm}^M$ and consistent features $\{\mathbf{C}^m\}_{m \neq dm}^M$. Then, after two levels of diversity learning, the diversity information of the remaining modalities relative to the dominant modality can be learned while removing redundant information.

Feature-level diversity learning can extract the diversity features between non-dominant and dominant modalities through mutual information and contrastive learning, compress redundant information and separate consistent information from diverse information. Cluster-level diversity learning ensures balanced distribution of each cluster through Uniform Distribution Constraint and mutual information, and enhances the expression of dominant modality from the clustering level. Therefore, the two are synergistically enhanced, enhancing low-level feature expression through feature-level learning, aligning high-level semantic structures through cluster-level learning, and cluster-level can guide feature-level learning through back-propagation to further promote clustering diversity information.

**Feature-level Diversity Learning** In order to learn the diversity information at the feature level, we propose the following objective function:

$$min \quad \mathcal{L}_1 = I(X^{dm}; D^{dm}) + \sum_{m \neq dm}^M I(X^m; Z^m) + \sum_{m \neq dm}^M I(Z^m; D^{dm}). \tag{4}$$

Where $I(;)$ is the mutual information measurement. The first two terms describe the compression loss of the raw modalities after passing through the encoders, and the third term aims to learn the diversity information of the dominant modality. Inspired by the Information Bottleneck (IB) theory [24, 25], according to Eq. (4), this method can effectively screen out the minimum sufficient representation that is most critical to the subsequent tasks of each modality under the guidance of the dominant modality, thereby suppressing redundant modal noise and highlighting the diversity information between the modalities.

In order to further effectively strip away the potential diversity information in the non-dominant modality, the dominant modality $\mathbf{D}^{dm}$ is compared and aligned with the consistency features $\{\mathbf{C}^m\}_{m\neq dm}^{M}$ of the non-dominant modality. Since the encoder network on which the consistency features $\{\mathbf{C}^m\}_{m\neq dm}^{M}$ and the diversity features $\{\mathbf{Z}^m\}_{m\neq dm}^{M}$ rely share parameters, the alignment process can more accurately decouple and extract the correct diversity features from the non-dominant modality while maintaining the consistency semantics, thereby improving the expressiveness of the dominant modality. The contrastive loss between the dominant modality $\mathbf{D}^{dm}$ and the consistency features $\mathbf{C}^m$ is calculated as follows:

$$\ell^{dm,m} = -\frac{1}{N}\sum_{i=1}^{N}\log\frac{e^{s\left(d_i^{dm},c_i^m\right)/\tau_1}}{\sum_{s'\in Neg\left(d_i^{dm},c_i^m\right)}e^{s'/\tau_1}}, \quad s_{ij}^{dm,m}\left(d_i^{dm},c_j^m\right) = \frac{\left(d_i^{dm}\right)^T c_j^m}{\left\|d_i^{dm}\right\|\cdot\left\|c_j^m\right\|}, \quad (5)$$

where $\tau_1$ is a temperature hyperparameter, $Neg\left(d_i^{dm},c_i^m\right)$ represents the similarity set between negative sample pairs, $d_i^{dm}$ and $c_j^m$ represent the $i$-th and $j$-th samples of features $\mathbf{D}^{dm}$ and $\mathbf{C}^m$. Contrastive learning is introduced between the dominant modality and other modalities to facilitate the screening of consistent features and further improve the efficiency of mining diverse information. The common way to calculate contrast loss is to accumulate the contrast loss between all modalities, so the contrast loss function in this section can be expressed as:

$$\mathcal{L}_2 = \frac{1}{2}\Big(\sum_{m\neq dm}^{M}\ell^{dm,m} + \sum_{m\neq dm}^{M}\ell^{m,dm}\Big), \quad (6)$$

to sum up, the loss function of the feature-level diversity learning is expressed as follows:

$$\mathcal{L}_{FDL} = \mathcal{L}_1 + \mathcal{L}_2. \quad (7)$$

**Cluster-level Diversity Learning**  In order to fully capture the semantic coherence between the dominant modality and the non-dominant modality and improve the effectiveness of diversity features in clustering tasks, DDMC further introduces a diversity learning mechanism at the cluster level. Specifically, the diversity features extracted from the non-dominant modality and the dominant modality are input into independent cluster assignment layers to obtain the cluster assignment representation $\{\mathbf{A}^m\}_{m=1}^{M}$, so as to mine the diversity information between them in the cluster structure, thereby enhancing the clustering expression ability of the dominant modality. To this end, we designed the following objective function:

$$max \quad \mathcal{L}_3 = \sum_{m\neq dm}^{M} I(A^m;A^{dm}), \quad (8)$$

at the same time, in order to improve the stability and balance of the clustering results, we further apply the UDC method to adjust the cluster assignment representation $\{\mathbf{A}^m\}_{m=1}^{M}$ to obtain $\{\mathbf{A_U}^m\}_{m=1}^{M}$. The objective function of this item is:

$$min \quad \mathcal{L}_4 = \sum_{m}^{M} CE(A^m;A_U{}^m), \quad (9)$$

where $CE$ is the cross entropy [26]. The core of this method is to carry out standardization processing similar to Sinkhorn-Knopp [27]. This method can convert the soft probability prediction of each sample belonging to each cluster into a more balanced hard label assignment, so that each sample is assigned to only one cluster and the distribution of the number of samples in each cluster is kept as balanced as possible, thereby improving the discriminability and robustness of clustering. The specific implementation details are shown in Appendix A.1. Therefore, the overall objective loss function of cluster-level diversity learning $\mathcal{L}_{CDL}$ is:

$$\mathcal{L}_{CDL} = \mathcal{L}_4 - \mathcal{L}_3. \quad (10)$$

After being processed by the two-levels diversity learning module, the diversity information contained in the remaining modalities can be effectively extracted compared to the dominant modality.

---

**Algorithm 1** :Diversity-oriented Deep Multi-modal Clustering

---

**Input:** Multi-modal datasets $\{\mathbf{X}^m\}_{m=1}^M$; Number of clusters $K$; Trade-off parameters $\alpha$ and $\beta$;
Epoch number $E$; Temperature parameters $\tau_1$.
Initializing the network;
Select the dominant modality by Eq. (3);
**for** $i = e$ **to** $E$ **do**
    The dominant modality feature $\mathbf{D}^{dm}$, consistency features $\{\mathbf{C}^m\}_{m \neq dm}^M$ and diversity features
    $\{\mathbf{Z}^m\}_{m \neq dm}^M$ are obtained through modality specific encoders;
    The clustering results $\{\mathbf{A}^m\}_{m=1}^M$ of various modality are obtained through cluster layers;
    Calculate $\mathcal{L}_{FDL}$ with Eq. (29), Eq. (31), Eq. (6) and Eq. (7);
    Calculate $\mathcal{L}_{CDL}$ with Eq. (31), Eq. (9) and Eq. (10);
    Calculate $\mathcal{L}_{DDC}$ by Eq. (12);
    Optimize all parameters by minimizing Eq. (13);
**end for**
**Output:** Multi-modal clustering assignment $Q$.

---

## 2.4 Clustering Module

In order to alleviate the information interference and feature confusion problems that may be caused by direct modal fusion, this paper first enhances the dominant modality. The enhanced dominant modality is expressed as:

$$\mathbf{D}_{Enhanced}^{dm} = Concat(D, Z^1, ..., Z^m, ..., Z^M)(m \neq dm). \tag{11}$$

Then, the enhanced dominant mode is clustered by DDC to obtain the cluster assignment matrix $Q$. DDC [28] is an effective unsupervised clustering method in deep learning, which aims to improve clustering performance by optimizing the dissimilarity measure between sample distributions. It consists of three parts, the first part is intra-cluster compression and inter-cluster separation, the second part is the orthogonality constraint, and the third part is the assignment of simple simplex corners, the loss optimization function of the clustering module is as follows:

$$\mathcal{L}_{DDC} = \frac{1}{K} \sum_{i=1}^{K-1} \sum_{j>i} \frac{\delta_i^T \mathbf{K} \delta_j}{\sqrt{\delta_i^T \mathbf{K} \delta_i \delta_j^T \mathbf{K} \delta_j}} + \text{triu}(Q^T Q) + \frac{1}{K} \sum_{i=1}^{K-1} \sum_{j>i} \frac{\lambda_i^T \mathbf{K} \lambda_j}{\sqrt{\lambda_i^T \mathbf{K} \lambda_i \cdot \lambda_j^T \mathbf{K} \lambda_j}}. \tag{12}$$

Where $\mathbf{K}$ is the Gaussian matrix kernel, $\delta_i, \delta_j$ denotes the column vectors of the cluster assignment matrix $Q$, and $K$ is the number of clusters. $\lambda_i$ is the column vector of $B = [B_{ab}]$, defined as: $B_{ab} = \exp\left(-\|\alpha_a - e_b\|^2\right)$.

## 2.5 Optimization

In summary, DDMC is achieved by minimizing the following loss function:

$$\mathcal{L}_{total} = \mathcal{L}_{DDC} + \alpha \mathcal{L}_{FDL} + \beta \mathcal{L}_{CDL}. \tag{13}$$

Where $\alpha, \beta$ are trade-off parameters. To optimize the mutual information loss term in Eq. (4), we introduce a variational optimization strategy. Specifically, we estimate the lower bound of the mutual information term $\mathcal{L}_1$ through a variational method and use this as its optimization target, because maximizing its variational lower bound can achieve an unbiased estimate of $\mathcal{L}_1$. The specific implementation process is presented at Appendix A.2 and the training steps of this method are summarized in Algorithm 1.

## 2.6 Theoretical Analysis

**Theorem 1.** By optimizing Eq. (7), it is possible to remove the private redundant information of each modality while obtaining the diversity information of the remaining modality relative to the dominant modality.

**Proof:** For any three modalities ($m \in \{u, dm, v\}$), where $dm$ is the dominant modality. We introduce a simplified assumption: when the training is fully converged, the encoder can accurately capture the joint information. According to the data processing inequality, , the mutual information between representations is upper-bounded by that between the original modalities:

$$I(Z^u; D^{dm}) \leq I(X^u; X^{dm}), \quad I(C^u; D^{dm}) \leq I(X^u; X^{dm}), \tag{14}$$

because $Z^u \leftrightarrow X^u \leftrightarrow X^{dm} \leftrightarrow D^{dm}$ forms a Markov chain. Similarly,

$$I(Z^v; D^{dm}) \leq I(X^v; X^{dm}), \quad I(C^v; D^{dm}) \leq I(X^v; X^{dm}). \tag{15}$$

At best, the information that can be shared between each modality and the dominant modality satisfies:

$$
\begin{aligned}
I\left(Z^u; D^{dm}\right) + I\left(C^u; D^{dm}\right) &= I\left(X^u; X^{dm}\right), \\
I\left(Z^v; D^{dm}\right) + I\left(C^v; D^{dm}\right) &= I\left(X^v; X^{dm}\right).
\end{aligned}
\tag{16}
$$

Since the optimization objective tends to maximize $I(Z^u; D^{dm})$ and $I(C^u; D^{dm})$, while the compression loss enforces the representation to remain concise, the optimal strategy is to assign redundant shared information to a low-dimensional subspace. Therefore, we let the encoding satisfy:

$$I\left(C^u; D^{dm}\right) = I\left(X^u; X^{dm}\right), \quad I\left(Z^u; D^{dm}\right) = 0. \tag{17}$$

That is, $C^u$ carries all common information between $X^u$ and $X^{dm}$, while $Z^u$ is independent ofis independent of $D^{dm}$ and contains only modality-specific information. Similarly,

$$I\left(C^v; D^{dm}\right) = I\left(X^v; X^{dm}\right), \quad I\left(Z^v; D^{dm}\right) = 0. \tag{18}$$

At this time, $C^u, C^v$ and $D^{dm}$ share the redundant information between all modalities, while $Z^u, Z^v$ each only carries the remaining independent information. According to the chain rule of mutual information:

$$I\left(X^u; Z^u, D^{dm}\right) = I\left(X^u; D^{dm}\right) + I\left(X^u; Z^u | D^{dm}\right). \tag{19}$$

When $I(Z^u; D) = 0$, it follows that $I(X^u; Z^u | D^{dm}) = I(X^u; Z^u)$. Assuming complete information preservation, we have:

$$I(X^u; Z^u | D^{dm}) = I(X^u; Z^u), \quad I\left(X^u; Z^u, D^{dm}\right) = H\left(X^u\right), \tag{20}$$

where $H()$ is the entropy. Finally, we obtain:

$$I\left(Z^u; X^u\right) = H\left(X^u\right) - I\left(X^u; X^{dm}\right) = I\left(X^u; X^u | X^{dm}\right). \tag{21}$$

That is, $Z^u$ captures the amount of information that $X^u$ still has given $X^{dm}$, which is exactly the diversity information of $X^u$ relative to $X^{dm}$. The same is true for $Z^v$. The above derivation shows that introducing modal contrast loss and mutual information loss under the compression loss constraint of the variational autoencoder will decouple the latent variables of the non-dominant modality from the latent variables of the dominant modality: the shared information is reflected in $C^u, C^v$, while $Z^u, Z^v$ only contains the diversity information relative to the dominant modality.

## 3 Experiment

In this segment, we carry out a series of experiments to verify the efficacy of the framework we have proposed. For more experimental data and experimental details, please refer to Appendix A.3.

### 3.1 Datasets

We evaluate the effectiveness of our proposed method by employing five well-known datasets, including Caltech-3V, Caltech-4V, ESP-Game, Flickr and IAPR. The Caltech image dataset [29], which comprises 1440 samples distributed across 7 distinct classes, is available in three multi-modal variants. **Caltech-3V** incorporates features of Wavelet moments [30], CENsus TRansform hISTogram

Table 2: Clustering results in terms of ACC and NMI on the multi-modal datasets. (The **bold** and underline value are the best and second best result, respectively)

| Methods | Caltech-3V | | Caltech-4V | | ESP-Game | | Flickr | | IAPR | |
|---|---|---|---|---|---|---|---|---|---|---|
| | ACC | NMI | ACC | NMI | ACC | NMI | ACC | NMI | ACC | NMI |
| KM | 46.3 | 31.3 | 54.6 | 46.7 | 43.2 | 29.4 | 40.9 | 22.5 | 38.9 | 17.2 |
| Ncuts(TPAMI'00) | 42.6 | 25.4 | 67.8 | 47.6 | 41.0 | 25.9 | 48.4 | 26.1 | 41.9 | 18.9 |
| AmKM | 46.9 | 31.5 | 44.9 | 30.6 | 49.9 | 34.7 | 41.0 | 21.6 | 40.4 | 17.0 |
| AmNcuts(TPAMI'00) | 43.7 | 25.5 | 41.8 | 24.9 | 33.5 | 19.1 | 48.2 | 26.2 | 42.2 | 18.9 |
| CoregMVSC (NeurIPS'11) | 54.4 | 45.3 | 64.9 | 54.5 | 40.1 | 28.8 | 41.0 | 26.8 | 35.1 | 18.4 |
| RMKMC (IJCAI'13) | 59.5 | 49.4 | 65.5 | 60.3 | 44.7 | 29.7 | 42.3 | 23.4 | 36.4 | 15.9 |
| SwMC (IJCAI'17) | 30.2 | 23.1 | 43.7 | 44.2 | 43.7 | 44.2 | 34.3 | 34.5 | 30.2 | 23.1 |
| ONMSC (AAAI'20) | 58.2 | 56.8 | 62.3 | 66.1 | 17.1 | 18.1 | 30.6 | 16.4 | 21.6 | 11.1 |
| SMCMB (TBD'23) | 67.2 | 54.5 | 74.4 | 67.0 | 54.9 | 40.5 | 52.8 | 32.1 | 34.8 | 16.4 |
| EAMC (CVPR'20) | 38.9 | 21.4 | 29.6 | 16.5 | 27.1 | 6.5 | 30.5 | 9.1 | 37.1 | 16.4 |
| DEMVC (InfoSci'21) | 38.7 | 27.0 | 48.4 | 39.7 | 35.5 | 21.6 | 44.8 | 25.2 | 30.1 | 13.8 |
| SiMVC (CVPR'21) | 56.9 | 50.4 | 61.9 | 53.6 | 35.3 | 16.2 | 45.6 | 26.3 | 42.7 | 18.5 |
| CoMVC (CVPR'21) | 54.1 | 50.4 | 56.8 | 56.8 | 51.8 | 38.2 | 49.3 | 30.6 | 46.7 | 21.5 |
| MFLVC (CVPR'22) | 63.1 | 56.6 | 73.3 | 65.2 | 52.1 | 39.4 | 53.8 | 32.8 | 47.3 | 22.6 |
| SEM (NeurIPS'23) | 69.2 | 59.2 | 82.6 | 75.3 | 36.6 | 23.5 | 53.1 | 30.9 | 42.2 | 18.9 |
| DIVIDE (AAAI'24) | 60.9 | 53.8 | 64.3 | 57.9 | 46.5 | 27.0 | 52.3 | 33.5 | 45.6 | 23.0 |
| SCMVC (TMM'24) | 75.9 | 66.3 | 84.4 | 72.9 | 36.1 | 24.8 | 54.2 | 32.3 | 46.5 | 24.1 |
| SSLNMVC (TMM'25) | 64.4 | 58.3 | 82.1 | 72.8 | 44.8 | 32.3 | 51.2 | 33.0 | 46.4 | 24.0 |
| **DDMC** | **76.7** | **68.8** | **90.3** | **82.7** | **60.9** | **40.9** | **58.7** | **36.5** | **49.5** | **28.3** |
| **Ours vs BestCompared** | **0.8↑** | **2.5↑** | **5.9↑** | **7.4↑** | **6.0↑** | **0.4↑** | **4.5↑** | **3.0↑** | **2.2↑** | **4.2↑** |

(CENTRIST) [31] and Local Binary Pattern [32], treating each feature type as a separate modality. **Caltech-4V** builds upon Caltech-3V by introducing an additional feature, namely Generalized Search Trees [33]. **ESP-Game** [34] dataset is derived from an online image tagging game, containing 11,032 images across 7 categories, with each sample having three modality descriptions.

Table 1: Details about the Multi-modal Datasets

| Dataset | Samples | Clusters | Dimension |
|---|---|---|---|
| Caltech-3V | 1440 | 7 | 40/254/928 |
| Caltech-4V | 1440 | 7 | 40/254/928/512 |
| ESP-Game | 11032 | 7 | 300/300/300 |
| Flickr | 12154 | 6 | 100/100/100 |
| IAPR | 7855 | 6 | 100/100 |

**Flickr** [22] dataset is a widely used multi-modal dataset for image retrieval, containing 12154 sample across 6 categories. It utilizes the same three modalities as ESP-Game. **IAPR** [35] is a comprehensive multi-modal image dataset, containing 6 categories with 7,855 samples and two features.

### 3.2   Compared Methods

To further substantiate the merits of our proposed approach, we conducted comparative experiments with a comprehensive set of eighteen baseline methods, classified into three types: **Single-Modal Clustering Methods:** K-Means(KM), Normalized Cuts(Ncuts) [36], All-modalities KMeans(AvKM), All-modal Normalized Cuts(AvNcuts); **Traditional Multi-modal Clustering Methods:** CoregMVSC [37], RMKMC [38], SwMC [39], ONMSC [40], SMCMB [41]; **Deep Multi-modal Clustering Methods:** MVSCN [42], DEMVC [43], SiMVC [44], CoMVC [44], MFLVC [45], SEM [20], DIVIDE [46], SCMVC [47], SSLNMVC [48]. All comparison method codes are obtained from the original authors through their official GitHub repositories or personal homepages.

**Evaluation Metrics**   The clustering performance is meticulously assessed by employing two extensively utilized metrics: Accuracy (ACC) and Normalized Mutual Information (NMI) [12]. [49] The higher the value of these two indicators, the better the performance of our model in terms of clustering accuracy and consistency.

### 3.3   Clustering Performance Analysis

Table 2 demonstrates the clustering effectiveness of our proposed method on five publicly available datasets, with the clustering outcomes in terms of ACC and NMI being reported. It is observed that

Table 3: Ablation Study on different multi-modal datasets (The **bold** and underline value are the best and second best result, respectively).

| Methods | Caltech-3V | | Caltech-4V | | ESP-Game | | Flickr | | IAPR | |
|---|---|---|---|---|---|---|---|---|---|---|
| | ACC | NMI | ACC | NMI | ACC | NMI | ACC | NMI | ACC | NMI |
| (1) $\mathcal{L}_{DDC}$ | 70.2 | 53.2 | 77.8 | 69.6 | 35.3 | 14.4 | 49.0 | 30.7 | 39.1 | 18.0 |
| (2) $\mathcal{L}_{DDC} + \mathcal{L}_{FDL}$ | 71.4 | 61.3 | 79.4 | 73.5 | 54.9 | 35.5 | 55.2 | 35.4 | 48.8 | 28.1 |
| (3) $\mathcal{L}_{DDC} + \mathcal{L}_{CDL}$ | 75.4 | 64.3 | 82.1 | 81.8 | 36.6 | 22.0 | 55.3 | 36.3 | 45.4 | 24.0 |
| (4) All Modules (The Proposed Method) | **76.7** | **68.8** | **90.3** | **82.7** | **60.9** | **40.9** | **58.7** | **36.5** | **49.5** | **28.3** |

our method significantly outperforms the compared single/all-view, traditional and deep multi-modal clustering methods. The remarkable results demonstrate the strong clustering capability of our method, which is attributed to the two levels diversity learning that is capable of acquiring diverse knowledge and eliminating redundant information.

The proposed method in this study demonstrates remarkable adaptability. On the Caltech dataset, as the number of modalities increases, the ACC and NMI of the model show a steady upward trend, owing to the rich complementary information contained in multi-modal data. As the number of modalities increases, the model can obtain data features from more dimensions. This indicates that the proposed method is capable of efficiently integrating information from different modalities and fully leveraging the unique features and clustering information contained in each modality, thereby significantly enhancing the accuracy and consistency of clustering. However, such an expected improvement in clustering performance is conspicuously absent when observing the outcomes of the EAMC and DEMC methods. These findings indicate that these methods might not be sufficiently robust when confronted with variations in the number of modalities. Compared with the best competitors, our method also obtains excellent performance. For example, on the ESP-Game dataset, our method demonstrates a substantial improvement in both ACC and NMI compared to the second-best method (SMCMB), achieving increases of 6.0% and 0.5% respectively. In terms of ACC and NMI results on the five tested multi-modality datasets, the proposed method achieves average improvements of 3.88% and 3.52% over the second-best comparison method, respectively. This highlights the stability of our method across different datasets.

### 3.4 Ablation Study

In this part, we give an ablation study to further show the effectiveness of different components of our method. The DDMC includes a clustering module and two diversity learning modules, which means there are four possible combinations. The outcomes of these combinations are presented in Table 3. The results indicate that the performance generally improves as more modules are incorporated into the model. For instance, on the Caltech-3V dataset, the baseline model achieved an ACC of 70.2% and a NMI of 53.2%. In contrast, the model that includes all modules saw a significant enhancement in performance, with an ACC of 76.7% and a NMI of 68.8%. Similar trends are observed across other datasets, with particularly notable performance enhancements in the Caltech-4V and IAPR datasets when all modules are integrated into the model. Specifically, the model equipped with all modules achieved an ACC of 90.3% and a NMI of 82.7% on the Caltech-4V dataset, and an ACC of 49.5% and a NMI of 28.3% on the IAPR dataset. These results underscore the superior performance of the model when all components are utilized.

### 3.5 Enhancement $vs$ Fusion

In addition, we replaced the final enhancement step with the commonly used modality fusion strategy, and the comparison results are shown in Figure 4. Experiments show that the enhancement strategy outperforms the fusion method on all datasets; especially on the ESP-Game dataset, the ACC is improved by about 11%. This result shows that compared with modality fusion, dominant modality enhancement can significantly improve clustering performance, while the noise introduced in the fusion process will weaken the role of the dominant modality and reduce the clustering effect, further verifying the effectiveness of the DDMC.

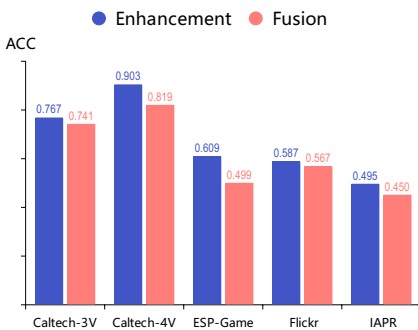

Figure 4: Enhancement $vs$ Fusion

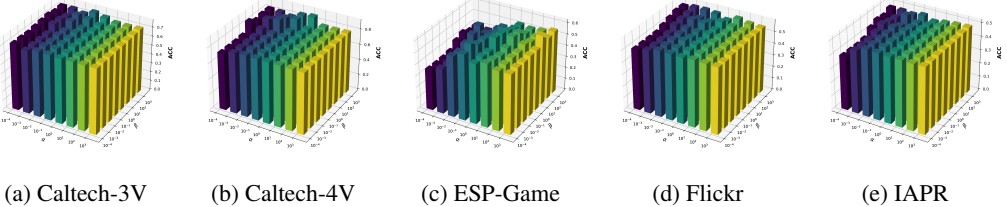

| (a) Caltech-3V | (b) Caltech-4V | (c) ESP-Game | (d) Flickr | (e) IAPR |

Figure 5: Parameter analysis of DDMC on different datasets.

## 3.6 Parameter Sensitivity

In the proposed method, we use two trade-off parameters $\alpha$, $\beta$ to balance $\mathcal{L}_{FDL}$, $\mathcal{L}_{CDL}$. To thoroughly investigate the sensitivity of these parameters, we conducted extensive experiments on all datasets with various parameter configurations. Specifically, we utilized a grid search strategy to optimize the values of $\alpha$ and $\beta$ within the range $[10^{-4}, 10^{-3}, 10^{-2}, 10^{-1}, 10^0, 10^1, 10^2, 10^3]$. The clustering ACC results are illustrated in Figure 5. As can be seen, our method exhibits robust performance across all datasets, with minimal performance degradation under most parameter settings. This suggests that our method is relatively insensitive to parameter fluctuations and maintains stable performance overall. Therefore, DDMC can have good performance in a large range of parameter values.

## 4 Conclusion and Limitations

This paper introduces a novel deep multi-modal clustering framework that leverages a dominant-modality enhancement strategy to mitigate noise from conventional feature-fusion. Rather than fusing all modalities indiscriminately, we identify the highest-quality modality as dominant, perform two-level diversity learning to extract diversity information from the remaining modalities, and augment the dominant modality accordingly. This strategy significantly improves the clustering performance, especially when the multi-modal data is unevenly distributed or has large quality differences.

In addition, our method is not without limitations. Although SI is an excellent indicator, there may still exist other theoretically feasible indicators. We will continue to study them in the future and introduce more robust dominant modality selection mechanisms. We also realize that this method performs poorly when dealing with incomplete modal data, especially when some modalities are missing or the data is unbalanced. In future research, We will explore more powerful selection strategies and effective strategies for extracting diversity information, such as combining multiple indicators or adaptive optimization methods, to ensure that the selection of the dominant modality is more convincing.

## Acknowledgment

The authors thank anonymous reviewers for their constructive comments. This work was supported by Henan Province Outstanding Youth Science Fund Program under Grant 252300421223, National Natural Science Foundation of China under Grant 62206254, 62325602, 62036010, and China Postdoctoral Science Foundation under Grant 2024T170843 and 2023M743186.

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

# A  Appendix

## A.1  The Details of Uniform Distribution Constraint

The role of UDC is to force the edge distribution of each cluster to be as close to uniform as possible when generating hard labels from the soft clustering distribution. Firstly: Firstly, perform exponentiation operations on the original cluster distribution matrix $\{\mathbf{A}^m\}_{m=1}^M$ to enhance the diagonal weights, Then, define two edge vectors $r \in \mathbb{R}^K$ and $c \in \mathbb{R}^N$. The goal is to make $P = \mathrm{diag}(r) A \, \mathrm{diag}(c)$ satisfy the following requirements:

$$P\mathbf{1}_K = \frac{1}{N}\mathbf{1}_N, \quad P^\top \mathbf{1}_N = \frac{1}{K}\mathbf{1}_K, \tag{22}$$

That is, $P$ approximates a 'coupling matrix' that is uniform for both rows and columns. Then iterative calculation is carried out through the Sinkhorn-Knopp algorithm:

$$r^{(t+1)} = \frac{\frac{1}{K}\mathbf{1}_K}{A^{(t)}}, c^{(t+1)} = \frac{\frac{1}{N}\mathbf{1}_N}{A^\top r^{(t+1)}} \tag{23}$$

Until convergence, transform the normalized matrix $P$ back to the $N \times K$ shape and take the maximum index of the $n$-th row to obtain the discrete label: $A_{Un} = \arg\max_k P_{n,k}$ The above process not only ensures the discreteness of the output labels, but also approximately satisfies the uniform distribution constraint of each cluster globally.

## A.2  Optimization

Taking a modality $X^1$ as an example, based on the definition of mutual information, we can get the following formula:

$$I\left(X^1; Z^1\right) = \int_{z^1, x^1} p\left(z^1, x^1\right) \log \frac{p\left(z^1, x^1\right)}{p\left(z^1\right)p\left(x^1\right)} = \int_{z^1, x^1} p\left(z^1, x^1\right) \log \frac{p\left(z^1 \mid x^1\right)}{p\left(z^1\right)}. \tag{24}$$

Since the posterior distribution $p\left(z^1, x^1\right)$ cannot be solved directly, another scalable distribution $q(z^1)$ is used to approximate $p(z^1)$. By minimizing the Kullback-Leibler (KL) divergence between the two distributions, $q(z^1)$ is optimized step by step to make it similar to $p(z^1)$. Specifically, since KL is non-negative, we can get:

$$\mathrm{KL}\left[p\left(z^1\right), q\left(z^1\right)\right] = \int_{z^1} p\left(z^1\right) \log \frac{p\left(z^1\right)}{q\left(z^1\right)} > 0$$
$$\Longrightarrow \int_{z^1} p\left(z^1\right) \log p\left(z^1\right) > \int_{z^1} p\left(z^1\right) \log q\left(z^1\right) \tag{25}$$

Now, Eq.(24) can be estimated as:

$$I\left(X^1; Z^1\right) < \int_{z^1, x^1} p\left(z^1, x^1\right) \log \frac{p\left(z^1|x^1\right)}{q\left(z^1\right)} < \int_{z^1, x^1} p\left(z^1|x^1\right) p(x^1) \log \frac{p\left(z^1|x^1\right)}{q\left(z^1\right)} \tag{26}$$

Since $\{\mathbf{Z}^m\}_{m \neq dm}^M$ does not contain the $dm$-th term and the dominant modality is not distinguished in the calculation of compression loss, for the sake of simplicity, we temporarily let $Z^{dm} = D^{dm}$, so that the compression loss in Eq.(4) can be estimated as:

$$\sum_{m \neq dm}^M I(X^m; Z^m) + I(X^{dm}; D) < \sum_m^M \int_{z^m, x^m} p(z^m|x^m) \log \frac{p(z^m|x^m)}{q(z^m)} \tag{27}$$

In order to remove unnecessary items, Monte Carlo sampling is used to replace $\sum_m^M p(x^m)$. Therefore, Eq.(27) can be further expressed as:

$$\sum_{m \neq dm}^M I(X^m; Z^m) + I(X^{dm}; D) < \frac{1}{N} \sum_i^N \{\sum_m^M \int_{z^m} p(z^m|x^m) \log \frac{p(z^m|x^m)}{q(z^m)}\} \tag{28}$$

where $N$ is the number of data samples. Assuming that $\sum_m^M p(z^m|x^m)$ conforms to the Gaussian distribution, the mean $\mu$ and variance $\sigma$ of each modality can be learned through the encoder. For the convenience of calculation, we reparameterize $\sum_m^M z^m$ by $\sum_m^M z^m = \mu(x^m) + \sigma(x^m) * \epsilon$, where $\epsilon$ is the standard normal distribution. Therefore, the objective loss function for calculating the compression loss in Eq.(4) can be expressed as:

$$
\sum_{m \neq dm}^{M} I(X^m; Z^m) + I(X^{dm}; D) < \frac{1}{N} \sum_i^N \{\sum_m^M \mathbb{E}_\epsilon \log \frac{p(z^m|x^m)}{q(z^m)}\}
$$
$$
\approx \frac{1}{N} \sum_i^N \mathbb{E}_\epsilon \{\sum_m^M KL[p(z^m|z^m), q(x^m)]\}
$$
(29)

where $\mathbb{E}_\epsilon$ represents mathematical expectation. For the calculation of diversity information loss in Eq.(4), we use the discrete joint probability estimation for calculation:

$$
\sum_{m \neq dm}^{M} I(Z^m; D^{dm}) = \sum_{m \neq dm}^{M} \sum_{i=1}^{d_m} \sum_{j=1}^{d_m} p^{m,dm}(i,j) \log \left( \frac{p^{m,dm}(i,j)}{p^m(i) p^{dm}(j)} \right)
$$
(30)

where:
$$
p^m(i) = \sum_{j=1}^{d_m} p^{m,dm}(i,j), p^{dm}(j) = \sum_{i=1}^{d_m} p^{m,dm}(i,j),
$$
$$
p^{m,dm}(i,j) = \frac{1}{2N} \sum_{n=1}^{N} \left[ Z_n^m(i) \cdot D_n^{dm}(j) + D_n^{dm}(i) \cdot Z_n^m(j) \right].
$$
(31)

Similarly, the calculation of cluster assignment mutual information $\mathcal{L}_3$ can also be optimized by Eq.(31).

## A.3 Further Analysis

**Evaluation Metrics**  ACC is used to evaluate the degree of correspondence between the assigned cluster labels and the true labels of the data points, essentially measuring the extent to which the clustering algorithm correctly identifies the inherent groupings within the dataset, which is defined as:
$$
ACC = \frac{\sum_{i=1}^{n} \delta(p_i, \mathrm{map}(q_i))}{n}
$$
(32)

where $n$ denotes the number of samples, $\delta(i,j) = 1$ if $i = j$ ($\delta(i,j) = 0, otherwise$), and $\mathrm{map}(q_i)$ represents the clustering result $q_i$ being matched to the ground truth $p_i$ through Hungarian algorithm[50]. NMI quantifies the mutual dependence between the clustering results and the ground truth, taking into account the distribution of data points across different clusters, formulated by:
$$
NMI(\Omega, C) = \frac{I(\Omega; C)}{(H(\Omega) + H(C))/2}
$$
(33)

where $\Omega$ indicates the original ground truth, and $C$ denotes the clustering information, $H()$ represents the entropy of the clustering result, and $I(;)$ indicates mutual information between clustering results.

**Details of the Compared Methods**  **K-Menas:** An unsupervised learning algorithm that divides data into K clusters so that the similarity of data points within clusters is high and the similarity between clusters is low.

**All-modal K-Means:** This algorithm concatenate all modal features and then executes KMenas algorithm

**Normailzed Cuts: [36]** An image segmentation algorithm based on graph theory is proposed to minimize the normalized cut value to optimize the image region partitioning, while considering the inter-group dissimilarity and intra-group similarity.

**All-modal Normalized Cuts:** The method uses Normalized Cuts for the joint representation of each data point to achieve clustering.

**CoregMVSC: [37]** A co-regularized multi-view spectral clustering method that enhance accuracy by enforcing clustering consistency across views.

**RMKMC: [38]** A multi-view multiple kernel clustering framework based on the restarted strategy and self-guiding mechanism.

**SwMC: [39]** A self-weighted mult-iview clustering algorithm that iteratively optimizes the target similarity matrix and view weights to effectively integrate multi-view data for clustering without requiring additional parameters.

**ONMMSC: [40]** A multi-view spectral clustering algorithm searches for the optimal matrix within the neighborhood of the linear combination of Laplacian matrices, breaking through the limitations of traditional methods to enhance clustering performance.

**SMCMB: [41]** This method mining rich information in multi-view data by joint learning of multiple bipartite graphs, and maintaining high efficiency on large-scale data sets, the time and space complexity is close to linear.

**EAMC: [10]** An end-to-end adversarial attention network that aligns potential feature distributions and quantifies modal importance, respectively, through adversarial learning and attention mechanisms.

**DEMVC: [43]** A multi-view clustering algorithm that utilizes common and complementary information from multiple views to achieve better clustering performance through deep embedded representation learning and collaborative training mechanisms.

**SiMVC: [44]** A deep multi-view clustering baseline model that does not require alignment of representation distributions

**CoMVC: [44]** A deep multi-view clustering algorithm that selectively aligns representations of different views at the sample level by contrasting learning frameworks.

**MFLVC: [45]** A multi-level feature learning framework is proposed, which can effectively reduce the interference of view private information by separating reconstruction target and consistency target.

**DIVIDE: [46]** A novel robust multi-view clustering method, which identifies global data pairs via high-order random walks and employs a decoupled contrastive learning framework to perform intra-view and inter-view contrastive learning in separate embedding spaces, thereby enhancing clustering performance and robustness against missing views.

**SSLNMVC: [48]** A deep multi-view clustering method that enhances the consistency of multi-view features through a consensus high-level feature learning module and aligns view-specific and view-consensus semantic labels using a self-supervised semantic calibration module.

**SEM: [20]** This method solves the representation degradation problem caused by contrast learning in multi-view scenarios through self-weighting and information reconstruction strategies.

**SCMVC: [47]** The method establishes a hierarchical feature fusion framework and a self-weighted contrastive fusion approach, effectively separating the consistency objective from the reconstruction objective.

**Implementation Details** Our experiments were conducted on a Windows 10 operating system, utilizing a powerful configuration equipped with 96 GB of system memory and a high-performance NVIDIA GeForce RTX 4090D GPU. We implemented the proposed framework using the PyTorch platform[51]. For all datasets, the training batch size was uniformly set to 512, and we utilized the Adam optimization algorithm with an initial learning rate of 0.0003. The configuration of parameters in the proposed model is detailed as follows. The hyperparameters $\alpha$ and $\beta$ are tuned to values ranging from 0.0001 to 1000, with each value being a power of 10. Given that 100 epochs proved to be ample for the training convergence of algorithm, we accordingly trained the model from the beginning up to 100 epochs. To enhance robustness and circumvent local minima, we trained the proposed model 10 times, reporting the clustering outcome with the minimal clustering loss. For all datasets, we utilized modal-specific variational encoders comprising three fully connected layers, each layer consists of a batch normalization layer and a RELU layer. The second layer and the output layer are set to 512. The clustering layer adopts a fully connected layer with a softmax layer. The parameter of the dropout layer is set to 0.1. The temperature hyperparameter in the comparative learning is set to 0.5.

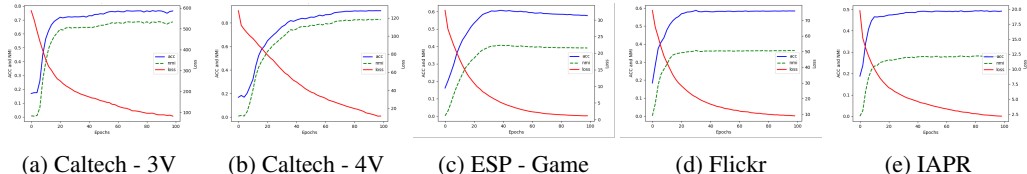

| (a) Caltech - 3V | (b) Caltech - 4V | (c) ESP - Game | (d) Flickr | (e) IAPR |

Figure 6: Convergence analysis of DDMC on the datasets in the order of Caltech - 3V Caltech - 4V, ESP - Game, Flickr and IAPR, respectively.

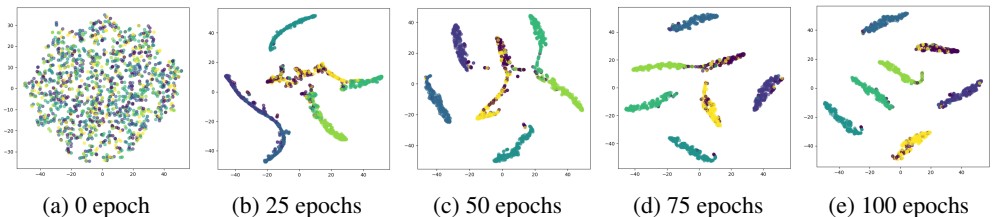

| (a) 0 epoch | (b) 25 epochs | (c) 50 epochs | (d) 75 epochs | (e) 100 epochs |

Figure 7: Evolution of cluster assignments during training on the Caltech-4V dataset.

**Convergence Analysis**    To evaluate the convergence properties of our proposed approach, we conducted a comprehensive series of experiments across a diverse range of datasets, including Caltech-3V, Caltech-4V, ESP-Game, Flickr, and IAPR datasets. As depicted in Figure 6, we present the objective loss values in conjunction with the clustering performance metrics, specifically ACC and NMI, throughout the training epochs. It is evident that the loss values experience a pronounced decline during the initial stages of training, particularly within the first 25 epochs. This rapid decrease indicates that the model is effectively learning and adjusting its parameters to minimize the loss function. Subsequently, the loss values begin to stabilize, suggesting that the model has reached a point of diminishing returns in terms of further loss reduction. The results consistently demonstrate that convergence for all metrics is achieved after the 100-th epoch threshold. This uniform convergence pattern across the diverse datasets highlights the robustness of our algorithmic iterations in attaining a balanced state, thereby ensuring consistent performance metrics irrespective of dataset variations.

**T-SNE Visualization Analysis**    In order to further substantiate the efficacy of our proposed frame-work, we employed the widely recognized t-sne tool to visualize the clustering outcomes of DDMC on the Caltech-4V dataset. The joint training of our approach yields satisfactory clustering results. To provide an intuitive depiction of the training evolution, we carried out t-sne visualizations of the clustering results at different training epochs, namely the 0-th, 25-th, 50-th, and 100-th epoch, as illustrated in Figure 7. In this figure, distinct colors are assigned to represent the various clusters generated by the DDC module. It is evident that as the number of training epochs increases, the clustering assignments tend to become more compact and well-separated from each other. This observation indicates the effectiveness of our proposed approach in enhancing clustering performance throughout the training process.

