# OpenReview forum: "Diversity-oriented Deep Multi-modal Clustering"
_NeurIPS.cc/2025/Conference — NeurIPS 2025 poster_

### Official Review · Reviewer_FbZy · 2025-06-25

**Clarity:** 3
**Significance:** 4
**Originality:** 4
**Rating:** 5
**Confidence:** 5

**Summary:**

To address the two challenges in most existing deep multi-modal clustering methods, the authors design a novel diversity-oriented multi-modal clustering method whose main idea is to enhance the dominant modality. They first select the dominant modality by prior knowledge or silhouette coefficient and then learn the diversity information between it and the other modalities. The resulting ACC and NMI values on various datasets show the superiority of the proposed method.

**Questions:**

See the strengths and weaknesses

**Ethical Concerns:**

["NO or VERY MINOR ethics concerns only"]

**Final Justification:**

The authors have well solved my concerns and I decide to raise the rating.

**Limitations:**

yes

**Paper Formatting Concerns:**

There exist no major formatting issues in this paper.

**Quality:**

3

**Strengths And Weaknesses:**

The proposed diversity-oriented multi-modal clustering method is novel, and the motivation is interesting which inspires the insights of dominant modality enhancement. The diversity learning and optimization method is described thoroughly, with a short but proper theoretical analysis, enhancing reader comprehension. The organization is quite clear and the experiment involving the enhancement or fusion is necessary to show the effectiveness of the diversity learning based method.

However, there are also few weaknesses in the manuscript.

a) To enhance the paper's quality, it is better to cite some latest multi-modal clustering references that reflect recent research developments.

b) What are the differences of the proposed method with existing methods except the diversity learning mechanism? The authors are suggested to give a clear description.

c) In section 2.2, the author should detail when the method needs use the SI to select the dominant modality.

d) The improvement in Table 2 is not clearly shown, and the authors should give a clear presentation.

e) Small issue: the authors are better to improve the paper format: e.g., the proof for the theoretical analysis in section 2.6 is given in one paragraph, which is not good to understand the process. It is better to split them into more paragraphs to organize them in a more readable fashion.

---

> ### Author Rebuttal · Authors · 2025-07-30
>
> Thank you very much for your careful review and precious suggestions. We attach great importance to the issues raised and have carefully revised and improved the problems such as the recent advances in multi-modal clustering, the details of the methodology and Overall article structure based on the comments. The following is our detailed response and explanation to each comment:
>
> Comment 1：To enhance the paper's quality, it is better to cite some latest multi-modal clustering references that reflect recent research developments.
>
> Thanks for your comment. In the field of deep multi-modal clustering, many excellent methods have also been proposed recently. These methods mainly enhance the performance of multi-modal clustering by optimizing the contrastive learning mechanism, improving the extraction accuracy of consistency information between modality, and effectively removing redundant information within modality. However, it has not effectively solved the problems of information conflicts and unfair weights brought about by modality fusion. We will elaborate in detail on multiple deep multi-modal clustering methods that can reflect the latest progress in the paper.
>
> Comment 2：What are the differences of the proposed method with existing methods except the diversity learning mechanism? The authors are suggested to give a clear description.
>
> Thanks for your comment. In addition to the diversity mechanism, this paper has two major differences from existing methods: (1) It uses the dominant modality-guided method for clustering. Compared with existing methods that consider all modalities at the same time and extract consistent information from them, it can effectively avoid the problem of information conflict between modalities. (2) There is no fusion throughout the process. Modal fusion will weaken the importance of high-quality modalities due to weight issues, while this method can well retain all useful information in high-quality modalities and enhance high-quality modalities through differential learning.
>
> Comment 3：In section 2.2, the author should detail when the method needs use the SI to select the dominant modality.
>
> Thanks for your comment. In our method, we give priority to using prior knowledge to select the dominant modality. This is because in many real-world tasks, domain experience can be used to quickly determine which modality is more representative and effective in expressing information. For example, "ESP-Game" is a dataset about online image annotation game, then vision is the most direct and efficient modality in annotation, and text plays the role of auxiliary recognition, which can help the model to better learn the detailed information in the picture. When there is a lack of clear prior judgment basis, we introduce the SI score as an auxiliary criterion and select the modality with a higher clustering SI score, that is, the modality with compact intra-class and good inter-class separation, as the dominant modality reference to enhance the robustness of the model in unsupervised situations.
>
> Comment 4：The improvement in Table 2 is not clearly shown, and the authors should give a clear presentation.
>
> Thanks for your comment. We will modify and optimize all the tables in the paper to ensure that the experimental results are clearer and more intuitive. These improvements aim to enhance the readability of the results and to more comprehensively demonstrate the performance gains and overall strengths of the proposed model across various evaluation metrics.  By improving the layout and clarity of the tables, we hope to make it easier for readers to interpret the experimental findings and appreciate the contributions of our method.
>
> Comment 5：Small issue: the authors are better to improve the paper format: e.g., the proof for the theoretical analysis in section 2.6 is given in one paragraph, which is not good to understand the process. It is better to split them into more paragraphs to organize them in a more readable fashion.
>
> Thanks for your comment. We will restructure the proof in Section 2.6 by breaking it down into logically organized and clearly separated paragraphs. This adjustment is intended to make the mathematical derivation more transparent and easier to follow.  Additionally, we will conduct a comprehensive review of the entire paper to further enhance its overall readability, including improvements in the clarity of explanations, formatting consistency, and presentation flow. These efforts aim to ensure that readers from diverse backgrounds can better understand and engage with the content of the paper.  Once again, we sincerely thanks for your valuable feedback.

---

> > ### Comment · Reviewer_FbZy · 2025-08-06
> >
> > Thanks for the author's rebuttal, which have addressed my conerns.  By the way, what does the 'clear prior judgment basis' mean in the rebuttal?

---

> ### Author Response · Authors · 2025-08-06
>
> Thanks for your comment. 'Clear prior judgment basis' refers to the prior knowledge to select the dominant modality, and this  knowledge comes from three aspects: 1) The modal priority in the dataset was clearly pointed out in previous work. In some multi-modal clustering methods, the paper clearly gives the weight priority of each modality. As in the experimental section of the existing work, the weights of each modal in the dataset used were sorted. 2) Textual descriptions or official introductions attached to the dataset. Multi-modal datasets may provide task background descriptions and modal composition explanations when released. For instance, dataset documents might indicate that image modality is the primary source of information in visual recognition tasks, or emphasize the significance of text modality in semantic understanding. 3) Prior knowledge in data similar to or related to the current task. If there is already multi-modal data that has been used in similar tasks or fields, the experience in choosing the dominant mode can be transferred to the current task as guidance. If there is no accurate grasp to select the dominant modality, we will use SI for auxiliary selection.

---

> > ### Comment · Reviewer_FbZy · 2025-08-09
> >
> > Thanks for the replies! The authors have well solved my concerns and I have no further questions!

---

### Official Review · Reviewer_85Ru · 2025-06-25

**Clarity:** 3
**Significance:** 3
**Originality:** 4
**Rating:** 5
**Confidence:** 5

**Summary:**

This paper introduces a deep multi-modal clustering framework based on dominant modality enhancement named DDMC. This approach aims to solve two challenges in the deep multi-modal clustering field, i.e., information conflicts between modalities and modality quality imbalance of many exiting multi-modal fusion and clustering methods. Experiments on different datasets validate the effectiveness of the proposed DDMC, also clearly demonstrating its feasibility and stability in multi-modal scenarios.

**Questions:**

(1) It seems that the proposed DDMC is the first work using dominant modality enhancement to solve the multi-modal clustering problem. Are there any possibilities for generalizing the idea for other related problems like multi-task or transfer learning.

(2) What are the sources of the compared methods in the experimental parts? They are reproduced by authors or given in the original paper? If it is from the corresponding papers, can the authors give the websites of them in the paper?

(3) It is good to see that the authors give the limitations of existing methods in two aspects, but it is unclear which the kind of related works the limitation corresponds to. It is recommended for the authors to give more details concerning the analysis in the

**Ethical Concerns:**

["NO or VERY MINOR ethics concerns only"]

**Final Justification:**

Thanks to the author for solving my concerns, and I decided to keep the original score.

**Limitations:**

yes

**Paper Formatting Concerns:**

There are no major formatting issues.

**Quality:**

4

**Strengths And Weaknesses:**

This paper proposes a deep multi-modal clustering framework based on dominant modality enhancement, and lots of experiments on different datasets validate the effectiveness of the proposed method, which reflects the stability in multi-modal clustering scenarios. Generally, this paper proposes a novel multi-modal clustering method, and the method introduction is clear, written in a clear and logic way. The general work flow is fluent and easy to understand. The effectiveness and feasibility of DDMC are clearly validated in the experiments from the main and appendix parts of the paper.

- Please find the specifics in Questions section.

---

> ### Author Rebuttal · Authors · 2025-07-30
>
> Thank you very much for your careful review and precious suggestions. We attach great importance to the issues raised and have carefully revised and improved the problems such as the application areas, the sources of the comparison methods and limitations of related work in the paper based on the comments. The following is our detailed response and explanation to each comment:
>
> Comment 1：It seems that the proposed DDMC is the first work using dominant modality enhancement to solve the multi-modal clustering problem. Are there any possibilities for generalizing the idea for other related problems like multi-task or transfer learning.
>
> Thanks for your comment. Your suggestion is an important inspiration for our future work. Multi-task learning aims to complete multiple related tasks simultaneously by sharing feature representations, but in practical applications, it often faces challenges such as task conflicts and unreasonable sharing methods. Drawing on the idea of "dominant modality guides diversity learning" in this study, tasks can be divided into dominant tasks and non-dominant tasks, thereby effectively alleviating conflicts and interference between tasks while enhancing the performance of dominant tasks. Transfer learning is committed to transferring the knowledge obtained in the source task to the target task, and often faces problems of negative transfer and large differences between the source/target domains. If combined with our method framework, by selecting the source domain with the richest information as the "dominant" and combining it with a differential guidance strategy for transfer, it is expected to improve the efficiency and stability of knowledge transfer. In future work, we will further explore the application potential and expansion methods of this idea in multi-task learning, transfer learning and other scenarios.
>
> Comment 2： What are the sources of the compared methods in the experimental parts? They are reproduced by authors or given in the original paper? If it is from the corresponding papers, can the authors give the websites of them in the paper?
>
> Thanks for your comment. The comparison methods used in the experimental section are all representative and high-performing approaches in the field of deep multimodal clustering. For fairness and reproducibility, we obtained the implementations directly from the authors’ official GitHub repositories, as referenced in their original papers. To facilitate future research and ensure transparency, we will clearly provide the URLs of these codebases in the revised version of the paper, so that other researchers can easily access and utilize them.
>
> Comment 3：It is good to see that the authors give the limitations of existing methods in two aspects, but it is unclear which the kind of related works the limitation corresponds to. It is recommended for the authors to give more details concerning the analysis.
>
> Thanks for your comment. First of all, almost all existing deep multi-modal clustering methods use modal fusion to find consistent information between modalities. All the methods mentioned in the related work use different levels of fusion, which corresponds to the limitations of multi-modal data conflicts and unfair fusion weights. For some of the few methods that do not use modal fusion to find cross-modal information, they also face the challenge of being unable to effectively resolve information conflicts. We will further supplement the relevant work in the article based on your suggestions.

---

> > ### Comment · Reviewer_85Ru · 2025-08-05
> >
> > Thanks for the author's reply, I have some other small questions:
> >
> > How would performance be affected if the dominant modality were randomly selected during the selection stage?
> >
> > In Section 3.5, the paper briefly mentions modality fusion strategies, which specific methods are adopted, average fusion, weighted fusion, or others？

---

> > > ### Author Response · Authors · 2025-08-06
> > >
> > > Thanks for the reviewer's comments. Our response are as follows:
> > >
> > > Comment1. How would performance be affected if the dominant modality were randomly selected during the selection stage?
> > >
> > > Thanks for the comment. If the dominant modality is randomly selected, the model performance will be reduced. Possible reasons are as follows: 1). The quality of the selected dominant modal information cannot be guaranteed. Diversity learning under dominant guidance will be built on an unreliable foundation and may instead introduce noise rather than enhance performance. 2). The selected dominant modality lacks of guidance. The dominant modality plays a core role in guiding the information extraction of the non-dominant modalities. If its own structure is not clear, it will lead to an unclear direction and poor alignment effect in the enhancement process.
> > >
> > > Comment2. In Section 3.5, the paper briefly mentions modality fusion strategies, which specific methods are adopted, average fusion, weighted fusion, or others？
> > >
> > > Thanks for the comment. In Section 3.5, we conducted a comparison of the results of enhanced dominant modalities and modality fusion. The experiments demonstrated the superiority of adopting the enhancement strategy. In the control experiment of modality fusion, we adopted the dynamic weighted fusion strategy. Specifically, these fusion weights can be learned during training. We select a set of non-normalized parameters as network parameters and train them in the optimization loss function. Finally, these parameters are passed through the softmax function to obtain the weights.

---

> > > > ### Comment · Reviewer_85Ru · 2025-08-09
> > > >
> > > > Thanks to the author for solving my concerns, and I decided to keep the original score.

---

### Official Review · Reviewer_a5KX · 2025-06-28

**Clarity:** 4
**Significance:** 3
**Originality:** 3
**Rating:** 5
**Confidence:** 3

**Summary:**

This work takes a different direction by learning diversity information between the dominant modality and other modalities. The proposed method first tackles information conflicts among modalities. Comprehensive experiments across multiple dimensions demonstrate the superior performance of this approach compared to existing state-of-the-art methods.

**Questions:**

Please refer to the weaknesses.

**Ethical Concerns:**

["NO or VERY MINOR ethics concerns only"]

**Final Justification:**

The authors have addressed my concerns, and I keeped my original score.

**Limitations:**

Yes

**Quality:**

3

**Strengths And Weaknesses:**

Strength:

S1: This paper introduces a novel multi-modal clustering framework, which uses a dominant-modality enhancement strategy to mitigate noise from conventional feature fusion.

S2: This strategy significantly improves the clustering performance when the data is unevenly distributed or has large quality differences.

Weaknesses:

W1: Technical Limitations. The authors propose two diversity learning paradigms: feature-level and cluster-level. What is the relationship between these paradigms? Do they operate through mutual enhancement, or how do they interact with each other?

W2: Methodological Issues. What is the original purpose of the silhouette coefficient? Given that there are various metrics similar to SI, could these alternative metrics also be applied for dominant modality selection? Please clarify the rationale behind choosing this specific metric.

W3: In Algorithm 1, the title 'Enhancement instead of fusion diversity learning for deep multi-modal clustering' appears inconsistent with the proposed method name. Additionally, the parameter description 'Number of categories of clusters K' is imprecise.

---

> ### Author Rebuttal · Authors · 2025-07-30
>
> Thank you very much for your careful review and precious suggestions. We attach great importance to the issues raised and have carefully revised and improved the problems such as the relationship between learning paradigms in the method, the selection of metrics, and the conflicts described in the article based on the comments. The following is our detailed response and explanation to each comment:
>
> Comment 1：Technical Limitations. The authors propose two diversity learning paradigms: feature-level and cluster-level. What is the relationship between these paradigms? Do they operate through mutual enhancement, or how do they interact with each other?
>
> Thanks for your comment. In our method, feature-level diversity learning can extract the difference features between non-dominant and dominant modalities through mutual information and contrastive learning, compress redundant information and separate consistent information from diverse information. Cluster-level diversity learning ensures balanced distribution of each cluster through Uniform Distribution Constraint and mutual information, and enhances the expression of dominant modality from the clustering level. Therefore, the two are synergistically enhanced, enhancing low-level feature expression through feature-level learning, aligning high-level semantic structures through cluster-level learning, and cluster-level can guide feature-level learning through back-propagation to further promote clustering diversity information.
>
> Comment 2： Methodological Issues. What is the original purpose of the silhouette coefficient? Given that there are various metrics similar to SI, could these alternative metrics also be applied for dominant modality selection? Please clarify the rationale behind choosing this specific metric.
>
> Thanks for your comment. Silhouette coefficient (SI) is an indicator used to evaluate the quality of clustering, which measures the closeness of data points in the same cluster and the degree of separation from other clusters. SI is usually used to evaluate the effectiveness of clustering methods such as K-means and hierarchical clustering. We also considered whether other indicators can replace SI, such as mutual information, reconstruction loss, Davies-Bouldin index (DBI) and other indicators, but none of them are suitable as indicators for selecting dominant modality. Specifically, the modality with the largest mutual information with the remaining modalities indicates that the modality has the highest correlation, and the modality with the smallest reconstruction error indicates that the information retention ability is the strongest. Neither of them can effectively select the modality with the richest information. DBI is easily disturbed by outliers and has poor stability. Therefore, SI is a suitable metric for selecting the dominant modality, as it can comprehensively consider intra-class compactness and inter-class separation, and also has good numerical stability and cross-mode comparison ability. In addition, there must be many theoretically feasible metrics. In our future work, we will conduct more extensive research to improve the selection strategy.
>
> Comment 3：In Algorithm 1, the title 'Enhancement instead of fusion diversity learning for deep multi-modal clustering' appears inconsistent with the proposed method name. Additionally, the parameter description 'Number of categories of clusters K' is imprecise.
>
> Thanks for your comment. We have noticed that the algorithm title deviates from the actual title and the description is inaccurate. We will conduct a detailed check of the full text in subsequent versions to ensure that the full text is consistent and the wording is correct. Thank you again for your careful review.

---

> > ### Comment · Reviewer_a5KX · 2025-08-06
> >
> > Thanks for the authors‘ reply.
> > Can the effectiveness of the two diversity learning paradigms be verified by experiments?

---

> > > ### Author Response · Authors · 2025-08-06
> > >
> > > Thanks for the reviewer's comment. The effectiveness of the two diversity learning paradigms can be experimentally verified. We verified the different paradigms by ablation study. The specific experimental results are presented in Table 3 of Section 3.4.  It is indicated that if neither of the two diversity learning paradigms is used (i.e., only through Deep Divergence based Clustering), the final clustering performance of the model is very poor. If only one learning paradigm is used, the clustering performance can be effectively improved. When both learning paradigms are used simultaneously, the clustering result is the best. Experiments fully demonstrate that the two diversity learning paradigms are synergistically enhanced and can effectively extract diversity information from different levels.

---

### Official Review · Reviewer_MBDR · 2025-07-01

**Clarity:** 4
**Significance:** 4
**Originality:** 4
**Rating:** 5
**Confidence:** 5

**Summary:**

In this paper a novel diversity-oriented deep multi-modal clustering method that focuses on dominant modality enhancement rather than modality fusion is proposed. The method first selects the modality with the highest average silhouette coefficient as the dominant modality and then leverages diversity learning to extract the differential information between the dominant modality and the remaining modalities. Experimental results show that the proposed method outperforms various compared single-modal and multi-modal methods.

**Questions:**

Please see the weaknesses for details.

**Ethical Concerns:**

["NO or VERY MINOR ethics concerns only"]

**Final Justification:**

Authors have addressed all my concerns.

**Limitations:**

yes

**Paper Formatting Concerns:**

No major formatting issues in this work.

**Quality:**

4

**Strengths And Weaknesses:**

Strengths: This paper proposes a novel diversity-oriented deep multi-modal clustering method that focuses on dominant modality enhancement rather than fusing different modalities. It introduces an effective and efficient method for multi-modal recognition community. In particular, the idea concentrating on improving dominant modality instead of fusing different modalities is quite different from the related works, which may be a new paradigm for exploring the correlated information among modalities. Additionally, many kinds of experiments including ablation study, parameter analysis and convergence analysis have been conducted to show the superiority of the proposed method.
Weakness: There are some weaknesses still exist in this paper listed in the following issues: 1) The authors mentioned that they decided the dominant modality method by prior knowledge or the Silhouette coefficient fraction, I am curious that what does the prior knowledge is. Can the authors give more details or information about the prior knowledge. 2) The font size in Figure 2/3 is too small, it is suggested to modify it to further improve the readability. 3) Are there any shortcomings of the proposed method? A more comprehensive analysis of the proposed method is needed.

---

> ### Author Rebuttal · Authors · 2025-07-30
>
> Thank you very much for your careful review and precious suggestions. We attach great importance to the raised issues and have carefully revised and improved the problems such as the incomplete method description, the small font size in the pictures and the inadequate description of the method's shortcomings based on the comments. The following is our detailed response and explanation to each comment:
>
> Comment 1：The authors mentioned that they decided the dominant modality method by prior knowledge or the Silhouette coefficient fraction, I am curious that what does the prior knowledge is. Can the authors give more details or information about the prior knowledge？
>
> Thanks for your comment. We give priority to using prior knowledge to determine the dominant modality in our method because in many practical tasks, we can quickly judge based on experience that the information of a certain modality is richer or more critical. For example, "ESP-Game" is a dataset about online image annotation games. Then, vision is the most direct and efficient modality in annotation, while text plays an auxiliary role in recognition, helping the model better learn the detailed information in the pictures. Therefore, selecting the modality with obvious advantages in this task as the dominant modality and integrating the diverse information of other modalities will help improve the overall learning effect. However, when the dominant modality cannot be clearly judged, we use the Silhouette coefficient score as an auxiliary basis and select the modality with high intra-cluster compactness and high inter-cluster separation as the dominant modality.
>
> Comment 2：The font size in Figure 2/3 is too small, it is suggested to modify it to further improve the readability.
>
> Thanks for your comment. We have noted the small font size issue in the figure, and we have enlarged the font size in this figure as well as all other figures throughout the paper to ensure better readability in both print and electronic formats.  In addition, we have carefully adjusted the proportion between the font size and graphical elements to achieve a more balanced and visually clear presentation.
>
> Comment 3：Are there any shortcomings of the proposed method? A more comprehensive analysis of the proposed method is needed.
>
> Thanks for your comment. The method proposed in this paper mainly improves the performance of multi-modal clustering by selecting the dominant modality and combining the diversity information of other modalities. However, our method still has the following shortcomings: (1) There is still room for improvement in the dominant modality selection strategy. Although SI is an excellent indicator, there may still exist other theoretically feasible indicators. We will continue to study them in the future and introduce more robust dominant modality selection mechanisms. (2) We also realized that this method performs poorly when dealing with incomplete modal data, especially when some modalities are missing or the data is unbalanced. We will add a clearer description of the limitations of this method in the paper.

---

> > ### Comment · Reviewer_MBDR · 2025-08-06
> >
> > Thanks for the author's response. It is good to see that the authors explain the prior knowledge can be obtained from experience with a dataset example given. Are there any other kind of prior knowledge?

---

> ### Author Response · Authors · 2025-08-06
>
> Thanks for your comment. Prior knowledge can not only be obtained from the experience of using the dataset, but also from the following aspects: 1) The textual description or introduction that comes with the dataset. Multi-modal datasets may provide task background descriptions, modal composition explanations, or usage suggestions when released. For instance, dataset documents might indicate that image modality is the primary source of information in visual recognition tasks, or emphasize the significance of text modality in semantic understanding. These descriptions can provide reasonable references for the priority of dominant modalities 2) Similar to or related to the current data prior knowledge. If there is already multi-modal data that has been used in similar tasks or fields, the experience in choosing the dominant modality can be transferred to the current task as guidance. For instance, in social media sentiment analysis, if previous studies have shown that the text modality is more direct and effective than the image modality in sentiment expression, then other datasets in this field can also prioritize the text modality as the dominant modality. 3) The conclusion given in the previous work. In some multi-modal clustering methods, the importance of each modality is explained. Just as the existing work which clearly presents the weight ranking of each modality in the used dataset. Of course, if there is no sufficient confidence to determine the dominant modality through prior knowledge, we will use SI for auxiliary selection.

---

### Decision · Program_Chairs · 2025-09-17

**Decision:**

Accept (poster)

**Comment:**

This paper proposes a diversity-oriented deep multi-modal clustering method in which the differences with existing methods lie in the dominant modality enhancement instead of modality fusion. Generally, all the reviewers reach a consensus that the proposed method is novel, with effectiveness validated by extensive experiments. Meanwhile, reviewers also raised some concerns in the initial review stage on the method description/working issues, algorithm analysis, generality to other fields, and experimental details. With the rebuttal provided by the authors, the reviewers suggest that the concerns have been well addressed, reaching final positive ratings. By carefully reading the paper, initial reviews, and rebuttal, I agree with the reviewers and recommend accepting this paper.